# Ergodic Archimedean dimers

Henrik Schou Røising [1▲], Zhao Zhang [2▼]

**1** Niels Bohr Institute, University of Copenhagen, DK-2200 Copenhagen, Denmark
▲ henrik.roising@nbi.ku.dk
**2** SISSA and INFN, Sezione di Trieste, via Bonomea 265, I-34136, Trieste, Italy
▼ zhao.zhang@su.se

July 10, 2023

## Abstract

We study perfect matchings, or close-packed dimer coverings, of finite sections of the eleven Archimedean lattices and give a constructive proof showing that any two perfect matchings can be transformed into each other using small sets of local ring-exchange moves. This result has direct consequences for formulating quantum dimer models with a resonating valence bond ground state, i.e., a superposition of all dimer coverings compatible with the boundary conditions. On five of the composite Archimedean lattices we supplement the sufficiency proof with translationally invariant reference configurations that prove the strict necessity of the sufficient terms with respect to ergodicity. We provide examples of and discuss frustration-free deformations of the quantum dimer models on two tripartite lattices.

# 1 Introduction

A perfect matching of a graph $G = (V, E)$ is a subset of edges $M \subset E$ such that every vertex in $V$ is adjacent to precisely one edge in $M$. Perfect matchings, known as close-packed dimer coverings in the condensed matter community, are widely studied in both graph theory and condensed matter physics. In statistical physics, there has been a long-standing interest in the properties of close-packed dimer coverings, such as their enumeration, dating back to Kasteleyn's exact results on the square lattice [1] and thermodynamically derived quantities [2, 3]. Moreover, the statistical properties of closed-packed dimer coverings appear in quantum dimer models at the so-called "Rokhsar–Kivelson" (RK) point at which such models become tractable. The square lattice quantum dimer model introduced by Rokhsar and Kivelson reads [4]:

$$H_{\text{RK}} = \sum_{p \in \text{plaquettes}} \left[ V \left( \left| \square \right\rangle \left\langle \square \right|_p + \left| \square \right\rangle \left\langle \square \right|_p \right) - J \left( \left| \square \right\rangle \left\langle \square \right|_p + \left| \square \right\rangle \left\langle \square \right|_p \right) \right]. \quad (1)$$

The diagonal term makes parallel nearest-neighbor dimers (thick red bonds) repel for $V > 0$, and the off-diagonal term flips the orientation of parallel nearest-neighbor dimers. At the RK point, defined as $V = J$, which is a critical point on the square lattice, quantum dimer models can be recast as a sum of projectors. With open boundaries the ground state is unique and formed by a uniform superposition of the classical dimer configurations [4, 5], which was first proposed to have physical relevance for the high-$T_c$ superconductors by posing a realization of Anderson's resonating valence bond (RVB) state [6, 7]. Since then, substantial progress has also been made towards technologies and ideas for realizing classical and quantum dimer models [8], with platforms ranging from arrays of ferromagnetic islands [9] to two-dimensional Rydberg atom arrays [10–14]. Quantum dimer models have also been putatively implemented on a programmable quantum simulator [13].

In the construction of quantum dimer models with the above-mentioned ground state property, ergodicity is often implicitly assumed: in order for all possible dimer coverings to be included in the ground state superposition, the local dimer terms included in the model Hamiltonian must generate the (dimer-constrained) space of classical configurations. The interplay between boundary effects, and the failure of ergodicity can lead to the phenomenon of (local) Hilbert space fragmentation [15, 16], which refers to the emergence of exponentially many dynamically disconnected subspaces. This terminology is further refined by how the disconnected Krylov subspaces—the distinct subspaces obtained by time-evolving various initial states—grow with system size [17, 18]. Furthermore, even when the boundary configuration is fixed, it is possible to modify the kinetic (off-diagonal) terms in the Hamiltonian to enforce a subset of the bulk configurations, those satisfying a certain condition, such as having non-negative height in height models, to be ergodic among themselves [19–24].

Ergodicity of perfect matchings for the square and the honeycomb lattice using one local term follows from the height function associated to the $U(1)$ Coulomb gauge symmetry [26]. In both these cases, a single ring-exchange dimer move, acting on the fundamental square and hexagonal plaquettes, respectively, is sufficient to ensure ergodicity[1], see Fig. 1. Kenyon and Rémila established that in the case of the triangular lattice three ring-exchange moves involving up to six triangular plaquettes are *sufficient* to ensure ergodicity of perfect matchings [25]. However, whether all three terms are *necessary* appeared less clear [5]. Among the 11 Archimedean lattices, which are edge-to-edge tilings of the plane consisting of regular polygons such that all vertices are identical under translations and rotations, the three above-mentioned lattices span the non-composite polygon tilings.

---

[1]To be precise, this statement is true up the exception of "staggered" configurations, which are configurations with no tiles where ring-exchange moves can be applied. These configurations will be further discussed in Sec. 3, and their existence is boundary dependent [15].

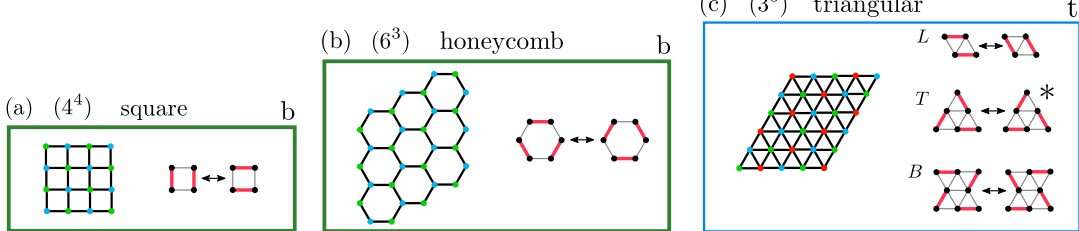

Figure 1: Finite sections of the three non-composite Archimedean lattices and a sufficient set of ring-exchange moves (up to rotations) to ensure ergodicity of close-packed dimer coverings [25]. In the lattice notation, e.g., "$(4^4)$", the base (resp. power) refers to the number of sides (resp. multiplicity) of the polygon encountered when going around a given vertex on the lattice. In (c) the moves have been dubbed $L$ ("lozenge"), $T$ ("triangle"), and $B$ ("butterfly"). On the square and the honeycomb lattice the sufficient move is also necessary with respect to ergodicity; they are marked with a green boundary. On the triangular lattice, marked with a blue boundary, the $T$ move ("∗") is proven redundant in Appendix A. Vertices are coloured to show that the square and honeycomb lattices are biparite (b), and the triangular lattice is tripartite (t).

Here, we formally establish sets of ring-exchange terms that are sufficient for ergodicity of perfect matchings on all the composite Archimedean lattices, using a constructive proof strategy inspired by Kenyon and Rémila. Moreover, by providing a set of reference configurations that prove the necessity of several ring-exchange terms, we establish on five of the eight composite Archimedean lattices that the sufficient terms are indeed all necessary for arbitrary boundary conditions. This result has direct applications to the formulation of quantum dimer models with a putatively extended RVB phase with topological order on a range of realizable non-bipartite lattices. Our approach can further be applied to, in principle, all polygon tilings with little additional effort.

## 2 Ergodicity of perfect matchings

In this section we prove a general strategy for connecting any two perfect matchings by a sufficient set of irreducible and local ring-exchange moves. The argument is constructive and as such provides a recipe for constructing the sufficient local moves to guarantee ergodicity. Note that "ergodicity" here is used as a synonym to "connectedness"; it refers to the property that any perfect matching can be transformed into any other. We will here consider open sections of a given lattice (not containing any holes) permitting a perfect matching. If the lattice is embedded on a topologically non-trivial surface with periodic boundary conditions, distinct non-contractible loops define winding sectors. Local ring-exchange terms can at most connect configurations within the same winding sector. With periodic boundary conditions imposed, there can also be "staggered configurations" that are frozen with respect to local terms, some of which are further discussed in Sec. 3 and Appendix B. We first summarize the main result.

### 2.1 Summary

Beyond the three non-composite Archimedean lattices and their set of ergodic ring-exchange moves shown in Fig. 1, we list in Fig. 2 the eight remaining and composite Archimedean lattices

with a sufficient set of ring-exchange moves (up to discrete rotations) to ensure ergodicity of perfect matchings. In Sec. 4 and Appendix B we supply these sufficiency results with examples showing how a series of ring-exchange moves are also necessary, allowing us to reach the conclusion that on the lattices marked with a green boundary in Fig. 2 the sufficient set of terms is also a necessary set of terms. An exhaustive and detailed example in the case of the

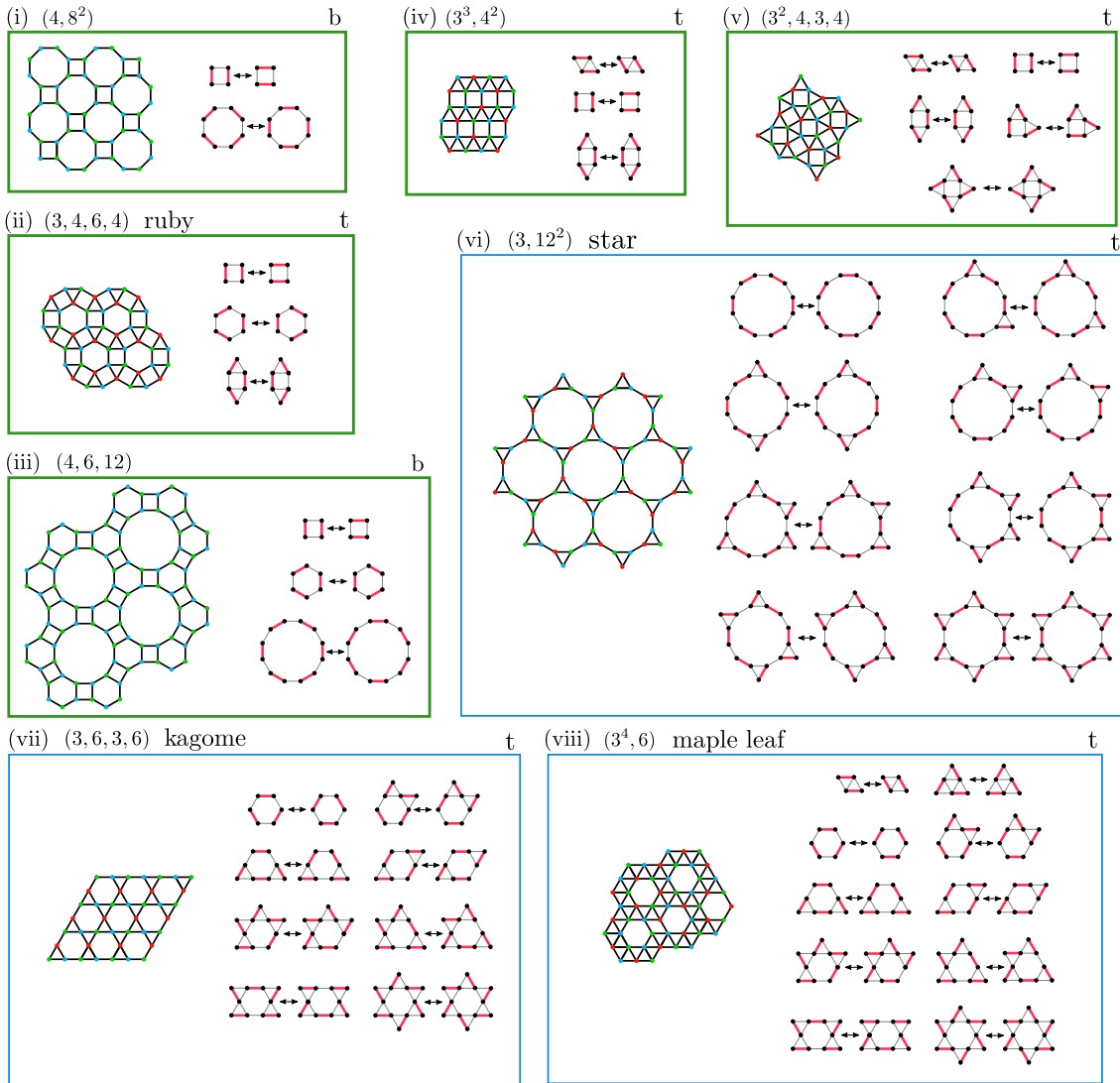

Figure 2: Finite sections of the eight composite Archimedean lattices and a set of sufficient ring-exchange moves (up to rotations) along irreducible cycles that ensure ergodicity of close-packed dimer coverings. On the lattices marked with a green boundary (i)-(v) the sufficient set of moves is also a necessary set of moves with respect to ergodicity, as we discuss in Sec. 4 and Appendix B.

kagome lattice is provided in Appendix C. We discuss implications of the results, as well as related and complementary results from the literature in Sec. 4.

## 2.2 Definitions and strategy

In Ref. [25] Kenyon and Rémila give a two-step argument to prove that any dimer covering, $M_1$, of a finite section of the triangular lattice without holes can be transformed into any other dimer covering, $M_2$, employing a set of the three local ring-exchange moves shown in Fig. 1(c).

The proof consists of making a series of transformations that changes $M_1$ into $M_1'$ and $M_2$ into $M_2'$ such that $M_1' = M_2'$. Here we generalize the principles in that paper and subsequently apply them to all the composite Archimedean lattices, although the idea in principle can be applied to a wider class of polygon tilings beyond the Archimedean ones.

We reiterate the observation from Ref. [25] that the *transition graph*, i.e., the union of two perfect matchings, $M_1 \cup M_2$, is a union of cycles, i.e., a subgraph where every vertex is touching two dimers, one from $M_1$ and one from $M_2$. A cycle will be called *trivial* if it has length 2, so that the two dimers from $M_1$ and $M_2$ overlap[2]. If a cycle is not enclosed by any other cycle, it will be said to be *maximal*. The two-step argument boils down to proving the two lemmas

**Lemma 2.1.** *A cycle in $M_1 \cup M_2$ not enclosing any other cycle can be transformed, using a set of local ring-exchange moves, to a set of trivial cycles.*

**Lemma 2.2.** *The cycles in $M_1 \cup M_2$ can be transformed, using a set of local ring-exchange moves, into a collection of cycles which are all maximal.*

Ergodicity then follows from two steps. First, if any cycle in the transition graph $M_1 \cup M_2$ is not maximal, and hence is enclosed by at least one cycle, we apply Lemma 2.2 until all cycles are maximal (the transformations on $M_1 \cup M_2$ is just a sequence of transformations acting on either $M_1$ or $M_2$). We note that in applying Lemma 2.2, Lemma 2.1 may have to be employed repeatedly; this is explained and illustrated in Sec. 2.4. When all cycles are maximal, no cycle can by definition contain any other cycle. Second, each maximal cycle is individually transformed into a set of trivial cycles using Lemma 2.1. The transformed transition graph has thus undergone local transformations that send $M_1 \mapsto M_1'$ and $M_2 \mapsto M_2'$ such that $M_1' = M_2'$. In Fig. 3 the second step in the above-mentioned procedure is illustrated on a finite section of the maple leaf lattice.

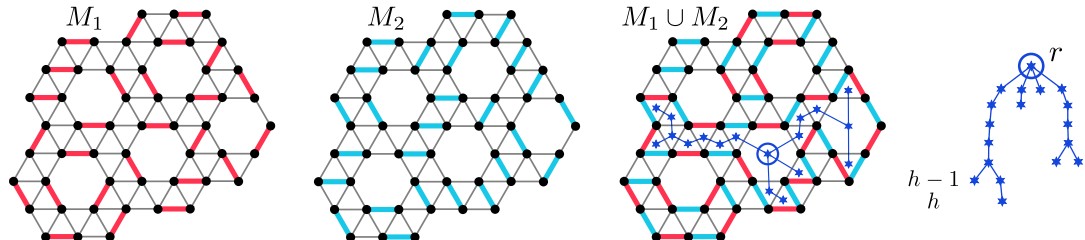

Figure 3: An example of two perfect matchings, $M_1$ (red bonds) and $M_2$ (blue bonds), on a finite section of the maple leaf lattice. To the right is the transition graph $M_1 \cup M_2$, which consists of cycles of varying length. A "6-ary" tree, where the centre of each plaquette contained in the cycle is marked by a blue star and connected to its neighbouring faces in the cycle, emerges for each non-trivial cycle that does not enclose any other cycle. As the root of the tree, $r$, we have here chosen an hexagonal face. The depth of the tree is denoted by $h$. Once the root is chosen, the iterative procedure in Lemma 2.1 acts with ring-exchange moves on plaquettes in vicinity to the node at level $h$ and shortens the tree until the entire cycle is transformed to a collection of trivial cycles.

The minimal set of moves relating perfect matchings of an Archimedean lattice in either a classical stochastic model or the dynamical (off-diagonal) terms of a quantum Hamiltonian consists of ring-exchange moves around what we call *irreducible even cycles* of the lattice. These are defined as cycles of even length that are otherwise chordless, but can contain chords that

---

[2]We note that if $M_1$ and $M_2$ are perfect matchings and $M_1 \cup M_2$ contains *no* trivial cycles, then $M_1 \cup M_2$ is known as a *fully packed loop* configuration.

join a pair of odd-length cycles. They are in other words irreducible in the sense that if it were to be further reduced to smaller cycles by any of its chords, it would inevitably result in an odd cycle. They can easily be identified by the definition, as they can not contain any even-edged face, and it suffices to include all tiles with an even number of triangles around every type of even-edged faces of the lattice. (On the triangular lattice, the role of even-edged face is played by a parallelogram consisting of two triangles, rendering the lozenge move ($L$) attaching no surrounding triangle, the triangle move ($T$) attaching two out of four surrounding ones, and the butterfly move ($B$) attaching all four, see Fig. 1(c).)

## 2.3 Proof of Lemma 2.1

Cycles in $M_1 \cup M_2$ are necessarily of even length as they emerge from two dimer coverings and the ring-exchange moves, that possibly are applied to merge and unravel loops in the first step, by construction are even-edged and hence preserve the even length property of all cycles acted upon. Since a proper candidate set of moves is exhaustive, any maximal cycle can be reduced to concatenations of cycles from the minimal set of irreducible ones, with the overlaps obeying a $\mathbb{Z}_2$ addition. A systematic way to bread down the maximal cycle into pieces is to consider the vertices of the dual lattice enclosed in the cycle. The dual lattice inside the cycle forms an "$n$-ary" tree; an example is shown for the maple leaf lattice in Fig. 3. Note that if there were a chordless cycle anywhere, there would be at least one vertex of the original lattice inside the cycle, which would either imply a monomer defect or contradict the assumption of the original cycle not containing any other cycles.

At first, let us exclude lattices containing triangles and extend the proof later to accommodate them. Within this restriction, we can start with any leaf node of the $n$-ary tree. Since each node has only one parent node, the face in the original lattice dual to a leaf node has only one edge not contained (uncovered by a dimer in either $M_1$ or $M_2$) in the cycle. This edge then cuts the cycle into a smaller and irreducible cycle. This implies the possibility of a ring-exchange move around that irreducible cycle in one of the two coverings that transforms the irreducible cycle into a pattern of trivial cycles and leaving the remaining non-trivial cycle shorter (now containing the previously uncovered edge).

For lattices containing triangular faces, but not adjacent ones, namely $(3,6,3,6)$, $(3,4,6,4)$, and $(3,12^2)$, we need to modify the definition of the nodes to exclude vertices dual to triangular faces and instead treat them as part of the branches. But this gives rise to the potential ambiguity of where to cut off the leaf node, as a branch passing through a triangular face crosses two edges, differing by whether the triangular face is to be included in the irreducible cycle or not. However, the length of the two would-be irreducible cycles differ by one, so only one of them is an irreducible even cycle. For the lattices $(3^3,4^2)$, $(3^2,4,3,4)$, and $(3^4,6)$ we count every second depth from the last parent node that is not dual to a triangular face as a node; these nodes will be cut off by one of the moves $L$ or $T$. And the even length of the irreducible cycle to be cut off also uniquely determines the shape of the resultant irreducible even cycle and the smaller remaining cycle. The procedure continues recursively until the cycle is broken down to a set of trivial cycles along the perimeter of the original cycle.

## 2.4 Proof of Lemma 2.2

To prove Lemma 2.2, which says that $M_1 \cup M_2$ can be transformed into a configuration in which no cycle contains another cycle, we just need to establish that cycles enclosing no other cycle but being enclosed by another cycle can be merged with the enclosing cycle. An illustration of the idea is shown in Fig. 4.

Starting from the innermost cycle containing no other cycles, we can apply Lemma 2.1 to transform it into a collection of trivial cycles. If there are additional cycles also enclosed by a

$M_1 \cup M_2$

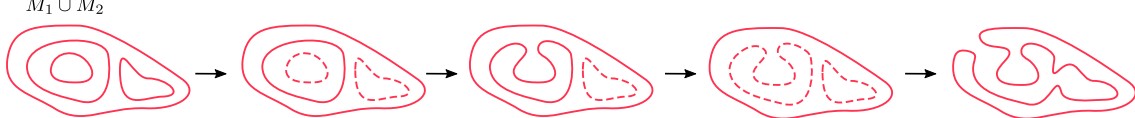

Figure 4: Illustration of the procedure in which all cycles in $M_1 \cup M_2$ are made maximal. Dashed lines here indicate a collection of trivial cycles, obtained after applying Lemma 2.1 to the innermost cycles that do not enclose other cycles.

common larger cycle, we apply Lemma 2.1 to them too, so that inside the common enclosing cycle there are only trivial cycles.

At the end of the above pre-processing step, all vertices inside of the enclosing cycle neighbouring the cycle are covered by one end of a trivial loop; the other vertex of the bond covered by this trivial cycle can either be neighbouring another vertex on the enclosing cycle (we label this case I) or another trivial cycle inside it (we label this case II). In case I, the union of edges between the involved four vertices, including a dimer on the enclosing cycle, defines an irreducible cycle on which a ring-exchange move can be used to merge the trivial cycle with the enclosing cycle (on lattices $(3, 6, 3, 6)$ and $(3, 12^2)$ the irreducible cycle will be of minimal length 6 and 12, respectively). In case II, one can trace the endpoint to another endpoint of a trivial cycle, whose other endpoint may or may not be neighbouring the enclosing cycle. If it does not, the tracing step can be continued by always following the direction closest in distance to the enclosing cycle, such that when an endpoint is neighbouring the enclosing cycle, an irreducible cycle must have formed, in which the number of uncovered bonds is odd. On either $M_1$ or $M_2$ one can therefore perform a ring-exchange move to absorb the trivial cycles in the enclosing one. This procedure repeats until no other trivial cycles are inside the enclosing one.

Once the "innermost" layer of cycles are untangled, we move on with the same procedure to the next level, and keep iterating "outwards" until all cycles are maximal.

## 3    Necessity and minimal sets of ring-exchange moves

In the preceding section we gave a constructive sufficiency proof for ergodicity of close-packed dimer configurations invoking small sets of ring-exchange terms. In this section we address the question of necessity of the sufficient terms for a subset of the Archimedean lattices (with a generic boundary of convex shape and of finite size greater than a few plaquettes[3]) by providing counterexamples of "staggered configurations" that would have formed disconnected Krylov subspaces if any of the moves from the sufficient sets are omitted. As explained below, we construct one such example for each of the moves on all the composite Archimedean lattices except $(3, 12^2)$, $(3, 6, 3, 6)$, and $(3^4, 6)$. Moreover, in the case of the triangular lattice, we show in Appendix A that the $T$ move (Fig. 1(c)) included in the sufficiency proof in Ref. [25] is redundant, by exhaustion of the local environments around a $T$ flippable tile. We caution that a candidate set of ring-exchange moves is not unique. However, when ring-exchange terms associated with the irreducible even-length cycles are considered, we are guaranteed that any other ring-exchange term one can think of can be reduced to compositions of those considered.

On the triangular lattice, the necessity of the $L$ and the $B$ move can be shown by appealing to particular configurations that only have these respective tiles flippable. For instance,

---

[3]In either of these two cases, the $T$ move in the triangular lattice might be necessary for ergodicity. On the other hand, for a particular shape of the boundary, the limited choices of perfect matchings compatible with the boundary might also render some of the generically necessary moves redundant.

the columnar configuration, see the left panel of Fig. 5, proves the necessity of the $L$ move. Necessity of the $B$ move can be exemplified by configurations with only flippable $B$ tiles, as shown in the right panel of Fig. 5. (For most of the neighborhood configurations surrounding a $B$-flippable tile, the $B$ move is decomposable into a sequence of $L$ moves. However, there are at least four configurations that are translations of the right panel of Fig. 5 that are frozen without the $B$ move.) In Ref. [27] the necessity of $L$ and $B$ in terms of ergodicity of dimer coverings was also discussed[4].

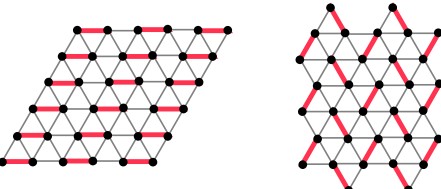

Figure 5: Left: columnar dimer configuration on the triangular lattice. Using this as a reference configuration shows that $L$ move is necessary for ergodicity. Right: a configuration proving the necessity of $B$ (cf. Refs. [27, 28]).

Following the same strategy, namely to construct translationally invariant configurations so that one does not need to worry about moves outside a depicted local region relating otherwise disconnected sectors, we identify further frozen configurations without certain types of ring-exchange moves in the minimal necessary set on a series of Archimedean lattices. Examples of columnar configurations, which prove the necessity of the simple square ring-exchange term, are shown in Fig. 6. A collection of further examples are listed in Appendix B. Taken together, these configurations allow us to conclude that on the Archimedean lattices $(4, 8^2)$, $(3^3, 4^2)$, $(3, 4, 6, 4)$, $(4, 6, 12)$, and $(3^2, 4, 3, 4)$ (marked with thick green boarders in Fig. 2), all the sufficient ring-exchange terms are indeed necessary to achieve ergodicity.

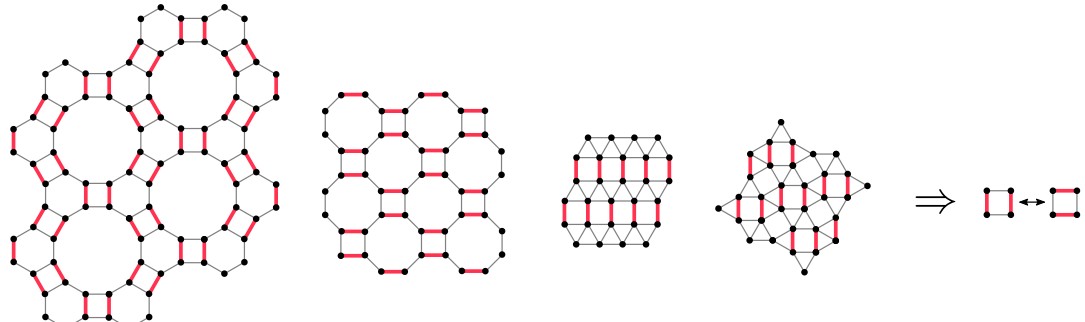

Figure 6: Necessity of the square ring-exchange move on lattices $(4, 6, 12)$, $(4, 8^2)$, $(3^3, 4^2)$, and $(3^2, 4, 3, 4)$ as implied by columnar reference configurations with only this type of tile flippable. Caveat: on the $(3^2, 4, 3, 4)$ lattice, the shown configuration has both square flippable and lozenge flippable tiles. However, since flipping the lozenges in this configuration does not result in any configuration other than the one in Fig. 9, the necessity of the square ring-exchange move follows when these two configurations are combined.

On the composite lattices $(3, 12^2)$, $(3, 6, 3, 6)$, and $(3^4, 6)$ the question of necessity is less clear for most ring-exchange terms, although some redundancy can be expected. While the

---

[4]In Ref. [27] evidence was mentioned that sectors of configurations are connected by lozenge moves alone, given common boundary conditions and nesting relations to a common "staggered configuration" similar to that of the right panel in Fig. 5.

sufficiency proof applies to both the $(3,6,3,6)$ kagome and the $(3,12^2)$ star lattice quantum dimer models, the ergodicity in these two cases is apparent since the set of moves collectively can bring all local configurations into each other [29]. Since the sufficient ring-exchange terms of Fig. 2 for these two lattices consist of one hexagonal and dodecahedral tile, respectively, and all its even-numbered decorations of triangles, a "pseudospin representation" of commuting kinetic operators can be formulated and ergodicity proven, along with exact enumeration results, see Refs. [29–31]. These commuting kinetic operators are sums of precisely the ring-exchange terms we find in Fig. 2 (vii) and (vi) around each hexagonal and dodecahedral tile on the kagome and star lattice, respectively.

## 4  Frustration-free quantum dimer models

Once a set of sufficient (with respect to ergodicity) and irreducible ring-exchange terms is established, an immediate application is the formulation of an ergodic quantum dimer model of which the ground state is the uniform superposition of all possible dimer coverings at the RK point, cf. the Rokhsar–Kivelson model in Eq. (1). Moreover, at the RK point the quantum dimer model becomes "frustration-free", which refers to the property that the ground state minimizes all the local (projector) terms of the Hamiltonian simultaneously. Ring-exchange terms naturally appear in perturbative expansions of the Hubbard model [32]. In this context we caution that the length of the irreducible cycles (in Fig. 2) dictate the order in which the corresponding ring-exchange term appear. Hence, finding a parent (Hubbard-like) Hamiltonian of an effective frustration-free quantum dimer model on the composite Archimedean lattices will require some level of fine tuning.

It is generally believed that the above-mentioned RVB ground state phase persists to an extended parameter region below ($V < J$) the RK point and has $\mathbb{Z}_2$ topological order on non-bipartite lattices [33]. On bipartite lattices, on the other hand, quantum dimer models only admit crystalline phases, and the RK point becomes a critical (gapless) point separating a plaquette phase ($V < J$) from an incommensurate staggered phase ($V > J$) [34, 35]. Moreover, non-bipartite lattices are generally expected to yield short-ranged dimer-dimer correlations [36], which can be contrasted to the square lattice case in which dimer-dimer correlations decay as $\sim 1/R^2$ [37]. Among the Archimedean lattices $(4^4)$, $(6^3)$, $(4,8^2)$, and $(4,6,12)$ are bipartite. The rest are non-bipartite, and, in fact, tripartite as we explicitly show with coloured tripartitions in Fig. 2. Below we provide two examples of frustration-free quantum dimer models on two tripartite lattices, given the results in Fig. 2.

On the ruby lattice, an ergodic quantum dimer model at the RK point is given by

$$
\begin{aligned}
H_{(3,4,6,4)} = J \sum_{p \in \text{plaquettes}} \Bigg[ & \left( \big| \hexlf \big\rangle_p - \big| \hexrt \big\rangle_p \right) \left( \big\langle \hexlf \big|_p - \big\langle \hexrt \big|_p \right) \\
& + \sum_{l=0}^{2} \left( \big| R^l_{\frac{2\pi}{3}} \rectlf \big\rangle_p - \big| R^l_{\frac{2\pi}{3}} \rectrt \big\rangle_p \right) \left( \big\langle R^l_{\frac{2\pi}{3}} \rectlf \big|_p - \big\langle R^l_{\frac{2\pi}{3}} \rectrt \big|_p \right) \\
& + \sum_{l=0}^{2} \left( \big| R^l_{\frac{2\pi}{3}} \diamlf \big\rangle_p - \big| R^l_{\frac{2\pi}{3}} \diamrt \big\rangle_p \right) \left( \big\langle R^l_{\frac{2\pi}{3}} \diamlf \big|_p - \big\langle R^l_{\frac{2\pi}{3}} \diamrt \big|_p \right) \Bigg],
\end{aligned}
\tag{2}
$$

where the sum over $p$ runs over the lattice plaquettes, and where $R_{\frac{2\pi}{3}}$ rotates the configuration counter-clockwise by $\frac{2\pi}{3}$. We note that in the quantum Hamiltonian of Ref. [38], the hexagonal ring-exchange term was left out. The configuration shown in Fig. 10 will form a disconnected Krylov subspace if the hexagonal term is omitted.

Returning to the triangular lattice, our above results imply that the $L$ and the $B$ moves

are necessary for ergodicity. We note that the Moessner–Sondhi model [5] only contains the $L$ move; however, it was conjectured in Ref. [5] that the addition of one four-dimer move would be enough to achieve ergodicity, which we here have formally proven true. Introducing deformation parameters $q_{L_l}$'s and $q_{B_l}$'s, an ergodic and $q$-deformed quantum dimer model can be formulated as

$$H_{(3^6)} = J \sum_{p \in \text{plaquettes}} \sum_{l=0}^{2} \left[ \frac{1}{\sqrt{1+q_{L_l}^2}} \left( \left| R_{\frac{2\pi}{3}}^l \text{⬠} \right\rangle_p - q_{L_l} \left| R_{\frac{2\pi}{3}}^l \text{⬠} \right\rangle_p \right) \left( \left\langle R_{\frac{2\pi}{3}}^l \text{⬠} \right|_p - q_{L_l} \left\langle R_{\frac{2\pi}{3}}^l \text{⬠} \right|_p \right) \right.$$
$$\left. + \frac{1}{\sqrt{1+q_{B_l}^2}} \left( \left| R_{\frac{2\pi}{3}}^l \text{⬠} \right\rangle_p - q_{B_l} \left| R_{\frac{2\pi}{3}}^l \text{⬠} \right\rangle_p \right) \left( \left\langle R_{\frac{2\pi}{3}}^l \text{⬠} \right|_p - q_{B_l} \left\langle R_{\frac{2\pi}{3}}^l \text{⬠} \right|_p \right) \right],$$
(3)

where we sum over all "lozenge" and "butterfly" plaquettes and their three orientations on the triangular lattice. Redundancies of the $T$ move on the triangular lattice in specific local

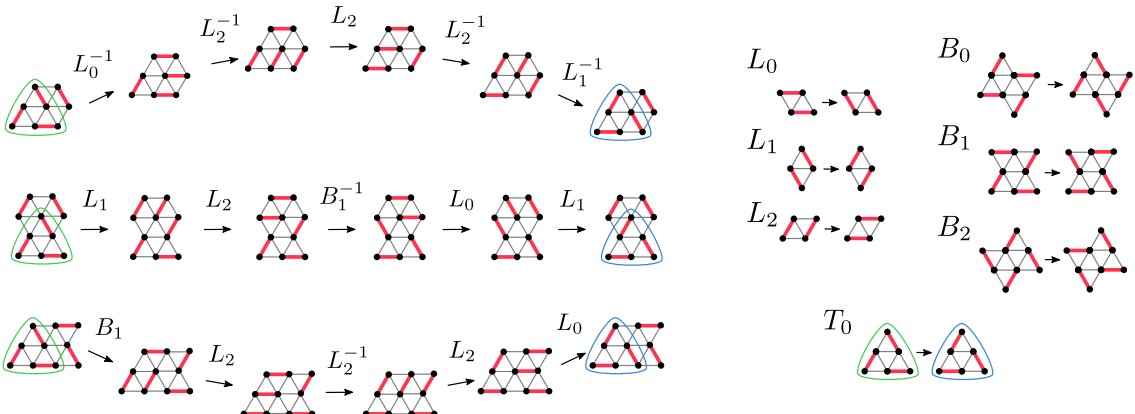

Figure 7: Examples of how the $T$ ring-exchange move (highlighted with green and blue boundaries) can be expressed in terms of $L$ and $B$ moves in specific local environments. These cocycle conditions impose consistency equations on the $q$-deformed Hamiltonian as discussed in Sec. 4.

environments, as exemplified by "pentagon relations" in Fig. 7, impose consistency relations on the deformation parameters $q_{L_l}$'s and $q_{B_l}$'s:

$$q_{L_1} q_{L_0} q_{B_1}^{-1} q_{L_2} q_{L_1} \equiv q_{L_1}^{-1} q_{L_2}^{-1} q_{L_0}^{-1} \equiv q_{L_0} q_{L_2} q_{B_1},$$
(4)

which permit the solutions

$$\left\{ \begin{array}{l} q_{L_0} q_{L_1} q_{L_2} = \pm 1, \pm i, \\ q_{B_j} = \pm q_{L_j}, \forall j = 0, 1, 2. \end{array} \right\}$$
(5)

Any of these parameter choices maintains a frustration-free Hamiltonian and a ground state being a weighted superposition of close-packed dimer coverings. When $q_{L_j} = q_{B_j} = 1$ for $j = 0, 1, 2$ the model is a Moessner–Sondhi model at the RK point with the addition of butterfly moves. Another rotationally invariant solution is $q_{L_0} = q_{L_1} = q_{L_2} = e^{\frac{2\pi i}{3}}$, such that $q_{L_0} q_{L_1} q_{L_2} = 1$ and $q_{B_j} = q_{L_j}$. Such a frustration-free deformation leads to a unique ground state as a superposition weighted by a phase depending on the numbers of dimers in each direction

$$|\text{GS}\rangle = \frac{1}{\sqrt{\mathcal{N}}} \sum_{M \in \text{dimer coverings}} e^{i\frac{\pi}{3} n_1(M) + i\frac{2\pi}{3} n_2(M)} |M\rangle,$$
(6)

where the normalization constant $\mathcal{N}$ is just the number of dimer coverings, as the weight is an imaginary number, $n_1$ denotes the number of dimers with an angle of $\frac{2\pi}{3}$ with respect to the horizontal and $n_2$ is the number of dimers with an angle of $\frac{4\pi}{3}$.

## 5    Conclusions and outlook

In this paper we have systematically mapped out sufficient sets of ring-exchange moves with respect to ergodicity of close-packed dimer coverings on all the 11 Archimedean lattices, among which 8 are non-bipartite and frustrated. We supplemented the constructive sufficiency proof with examples (by contradiction) of necessity of a series of moves, resulting in a minimal set of ergodic moves on 5 of the 8 composite lattices. These results have immediate applications in the formulation of quantum dimer models with a unique RVB ground state and topological order reminiscent to that of the $\mathbb{Z}_2$ Ising gauge theory [39,40], which in principle can be implemented and tested on near-term quantum simulators [13,41]. The ergodicity of the ring-exchange moves is crucial for making sure the numerical simulations, which usually employs worm algorithms [42] that update with ring-exchange on arbitrarily large cycles for efficiency, are indeed simulating the Hamiltonian, without mixing disconnected Krylov subspaces.

Among other polygon tilings it appears possible, based on the sufficiency argument given for some regular polygon tilings in this paper, that perfect (or maximal) matchings of quadrilateral tilings are ergodic under the Rokhsar–Kivelson ring-exchange move alone, again up to the possible exceptions of staggered configurations and distinct winding sectors. If true, this would cover all the 2D quasicrystals generated by the "de Bruijn grid method" [43,44]. As a caveat, however, we mention that in the Penrose tiling no perfect matchings exist [44], violating one of our basic assumptions. However, within the set of maximal matchings, which are dimer coverings with a minimal monomer density, ergodicity of the Rokhsar–Kivelson ring-exchange move alone appears plausible, though with putative fragmentation caused by impenetrable "monomer membranes" as explained in Ref. [44]. One could also consider ring-exchange moves involving both dimers and monomers around odd length cycles and study the ergodicity of these.

We note that there is an alternative route to proving ergodicity of close-packed dimer configurations on lattices permitting a well-defined height representation living on the dual lattice [26]. This strategy is concerned with showing that the height of a closed-packed dimer configuration always can be minimized, using local ring-exchange moves, until one of the minimal-height configurations is reached, as done in the case of 2-dimers in Ref. [45]. The height representation is also relevant with respect to formulating an appropriate field theory.

Future directions include attempting to generalize the construction to trimers [46] and $n$-mers, as well as ring-exchange moves that involve larger number of plaquettes due to additional constraints from longer range interactions [47]. Furthermore, it would be useful for characterizing the nature of Hilbert space fragmentation to determine the number of Krylov subspaces when a certain necessary move is absent [17]. In addition to the ergodicity breaking caused by the absence of necessary bulk moves in the current paper, and by the open boundary discussed in Ref. [15], one can also explore fragmentation due to defects in the lattice, which can both require additional moves and make some existing moves redundant, as the maple leaf lattice viewed as triangular lattice with defects has shown. In the mathematical literature the equivalent formulation of closed-packed dimer configurations as generalized domino tilings of the dual lattice [48] and the "arctic circle" phenomenon [49] have been extensively studied on the Aztec diamond. It is of interest to explore this phenomenon and the convergence of the arctic curve more broadly on the composite lattices considered here. This is especially interesting in light of the ergodic dimer moves established in this paper that translate to the

Glauber dynamics of the stochastic domino tilings.

Moreover, it can be mentioned that there are material candidates in which several lattices considered in this work have been or possibly can be realized. We caution, however, that this does not mean that quantum dimer models are necessarily relevant for the study of these materials. The superconducting $AV_3Sb_5$ ($A$: K, Rb, Cs) materials [50] realize the $(3, 6, 3, 6)$ kagome lattice, certain copper minerals may realize the $(3^4, 6)$ maple leaf lattice [51], the graphene allotrope graphenylene could realize the $(4, 6, 12)$ lattice [52], boron sheets may realize both the $(3^4, 6)$ and $(3^3, 4^2)$ lattices [53,54], and the $(3, 4, 6, 4)$ lattice could be realized by stacks of the topological insulator $Bi_{14}Rh_3I_9$ [55] to mention some.

Finally, we remark that the notion of "ergodicity" throughout this paper is used in the sense that the moves in the quantum Hamiltonian ensures the ground state of quantum model to be a superposition of all perfect matchings. It does not imply that the quantum Hamiltonian is ergodic in the eigenstate thermalization hypothesis satisfying sense, although on the square and triangular lattice this has been shown to be true, when restricted to each topological sector [56]. It would be interesting to extend those studies to the other Archimedean lattices.

# Acknowledgements

Conversations and correspondence with Matteo M. Wauters, Felix Flicker, Paul Fendley, Brian M. Andersen, Olav F. Syljuåsen, Roderich Moessner, Lev Vidmar, Bram Vanhecke, and Norbert Schuch are acknowledged. This work was supported by a research grant (40509) from VILLUM FONDEN.

# A  Decomposition of $T$ move in terms of $L$ and $B$ moves on the triangular lattice

In this Appendix we show how the $T$ move can be decomposed into $L$ and $B$ by either using the middle identity in Fig. 7 (referred to as I), or one of two presented below. In this Appendix we let $M$ denote a dimer configuration containing a $T$ tile, and refer to vertices in vicinity to this tile as labelled in Fig. 8. If $a_2b_3, a_3a_4 \in M$ (referred to as II), a $B$ move followed by

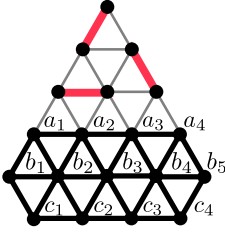

Figure 8: Labelled vertices in vicinity of a $T$ flippable tile. By exhaustion of configurations below the tile it is shown that $T$ can always be decomposed into $L$ and $B$.

four $L$ moves implements the $T$ move. By left-right symmetry, the $a_1a_2, a_3b_3 \in M$ scenario (referred to as III) is implemented by the reverse of the same sequence of moves. We show in the following that all the other possibilities can be brought to one of I, II, or III with some additional $L$ moves.

1. $a_2b_2, a_3b_4 \in M$: either $b_3c_2$ or $b_3c_3$ must be in $M$. Both cases reduce to I after two $L$ moves.

2. $a_2 b_2, a_3 b_3 \in M$: one $L$ move brings it to I.

3. $a_3 a_4, a_2 b_2 \in M$:

   (a) $b_3 b_4 \in M$: two $L$ moves brings to I.

   (b) $b_3 c_2 \in M$: an $L$ move brings to II.

   (c) $b_3 c_3 \in M$. Either $b_4 b_5 \in M$ or $b_4 c_4 \in M$. In both cases, three $L$ moves brings it to I.

4. $a_1 a_2, a_3 a_4 \in M$:

   (a) $b_2 b_3 \in M$: one $L$ move brings it to II.

   (b) $b_3 c_2 \in M$: either $b_2 b_1 \in M$ or $b_2 c_1 \in M$. In both cases, two $L$ moves brings it to II.

   (c) The other two cases are left-right symmetric with either (a) or (b) and can hence be brought to III with up to two $L$ moves.

5. All the other cases are left-right symmetric with respect to one of the above.

## B   Necessity of ring-exchange moves

We here provide examples of translationally invariant dimer configurations that have only one or a limited number of flippable tiles. These examples show (by contradiction) the strict necessity of several ring-exchange terms listed in Fig. 2. The example configurations are listed in Figs. 9-14.

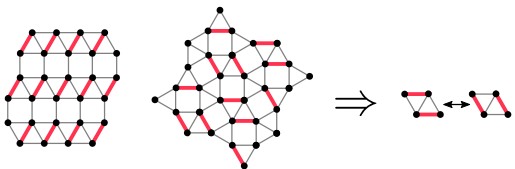

Figure 9: Necessity of the lozenge ring-exchange move on lattices $(3^3, 4^2)$, and $(3^2, 4, 3, 4)$, as implied by columnar reference configurations with only this type of tile flippable.

## C   Explicit sufficiency proof on the kagome lattice

To explicitly demonstrate the proofs of Lemmas 2.1 and 2.2 in Sec. 2.2, we here provide an exhaustive example on the kagome lattice. We note that this exhaustive strategy can be trivially extended to the arguably more complicated cases such as the maple leaf lattice. However, in the case of the maple leaf lattice one additionally has to invoke a series of composite ring-exchange moves (meaning that they can always be decomposed into the irreducible ones in Fig. 2(viii)), on top of the added combinatorial complexity regarding the shapes of the 6-ary tree. As mentioned in the main text, Refs. [29–31] offer alternative and economical ergodicity arguments on the $(3, 6, 3, 6)$ kagome and the $(3, 12^2)$ star lattice.

On the kagome lattice there are two distinct faces: hexagons and triangles. Corresponding nodes in the 6-ary tree formed by a cycle (not enclosing any other cycles) can have at most 6 and 3 children, respectively, leading to several possibilities for the shape of the 6-ary tree near its bottom, which we exhaust below. Reiterating the claimed result in Fig. 2(vii), a sufficient

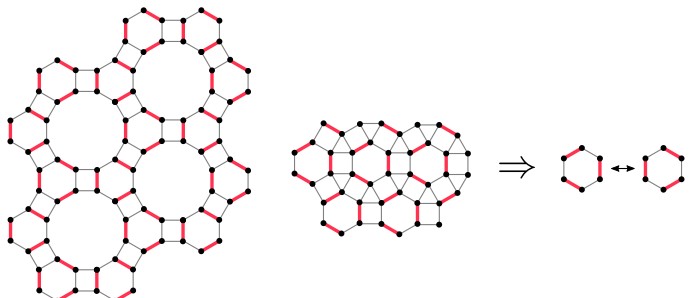

Figure 10: Necessity of the hexagonal ring-exchange move on lattices $(4, 6, 12)$, and $(3, 4, 6, 4)$, as implied by reference configurations with only this type of tile flippable. In the case of the $(3, 4, 6, 4)$ lattice, necessity of the square ring-exchange move (Fig. 6) is also implied by this reference configuration, since no triangular bond can be covered by employing the hexagonal ring-exchange move alone. In other words, flipping hexagons do not make any other tiles flippable without invoking the square ring-exchange moves.

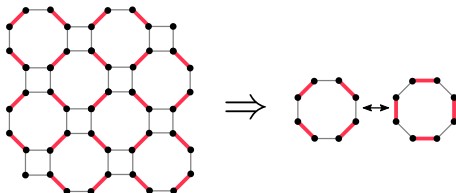

Figure 11: Necessity of the octagonal ring-exchange move on the $(4, 8^2)$ lattice, as implied by a reference configuration with only this type of tile flippable.

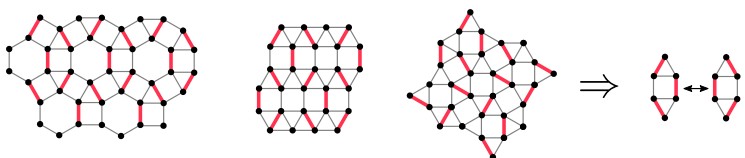

Figure 12: Necessity of the diamond-shaped ring-exchange move on lattices $(3, 4, 6, 4)$, $(3^3, 4^2)$, and $(3^2, 4, 3, 4)$, as implied by reference configurations with only this type of tile flippable.

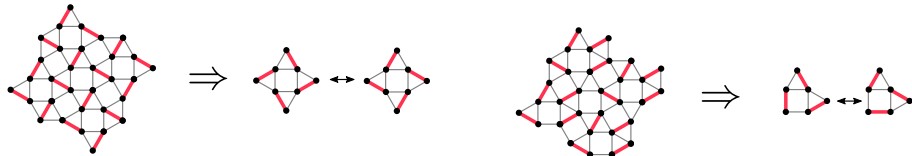

Figure 13: Necessity of the remaining two sufficient ring-exchange moves on the $(3^2, 4, 3, 4)$ lattice. Left: necessity of the star-shaped ring-exchange move is implied by a reference configuration with only this type of tile flippable. Right: necessity of the chair ring-exchange move is implied by a reference configuration with only chairs and lozenges flippable. Flipping the lozenges does not make any plaquettes except chairs flippable, and necessity of the chair move is thus implied when combined with Fig. 9.

set of irreducible ring-exchange moves on the kagome lattice, labelled here by $H_1$-$H_8$, is shown in Fig. 15.

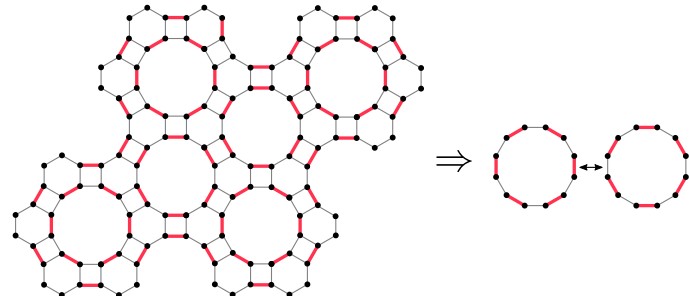

Figure 14: Necessity of the dodecagon ring-exchange move on the $(4, 6, 12)$ lattice, as implied by a reference configuration where only dodecagons and squares are flippable. Flipping the squares does not make any neighbouring plaquettes flippable, and necessity of the dodecagon move is thus implied when combined with Fig. 6.

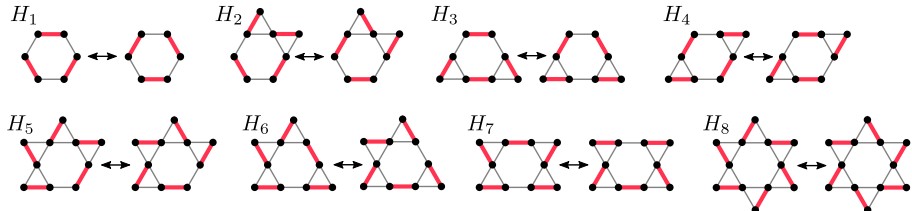

Figure 15: Ring-exchange moves, up to rotations of $\pi/3$ and $2\pi/3$, sufficient to achieve ergodicity of perfect matchings on the kagome lattice.

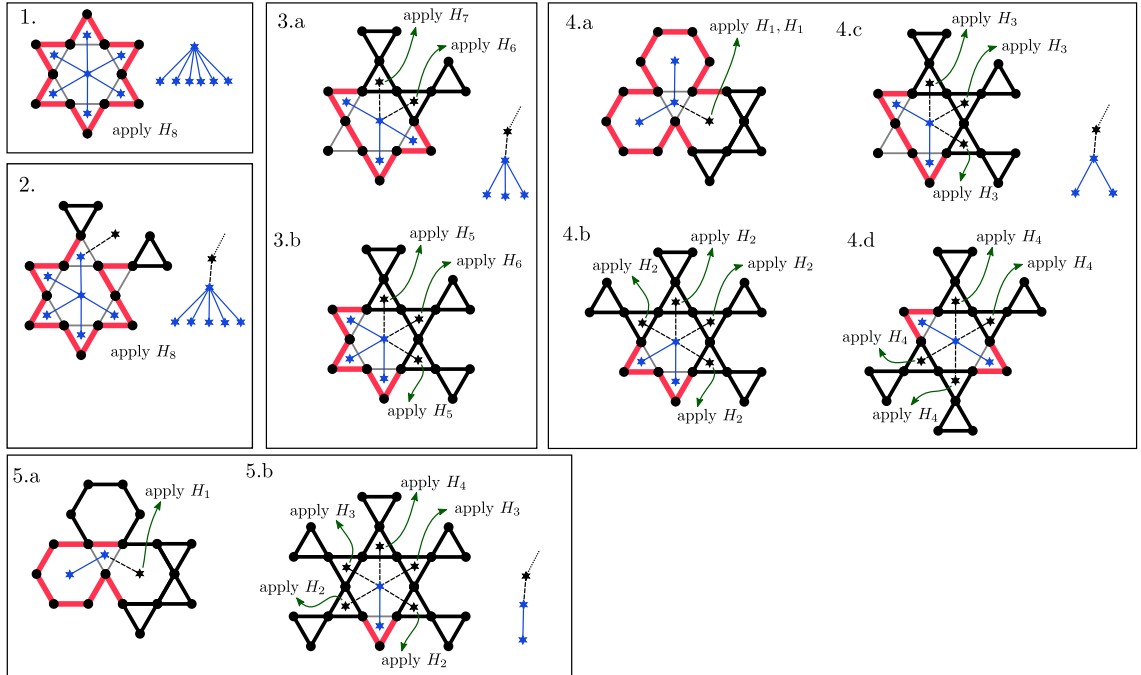

Figure 16: Exhaustion of possible cycle shapes in $M_1 \cup M_2$ near the bottom of the "6-ary" tree, shown to the right in each case. The black stars (and black dashed lines) indicate possible locations of the parent of node $A$ (at depth $h-1$). In the cases with multiple possibilities, the operation needed to shorten the "6-ary" tree is indicated by the green arrows (but not on the face at which the operation should be applied).

*Proof.* To prove Lemma 2.1 on the kagome lattice by exhaustion, we enumerate all possible shapes near the bottom of the 6-ary tree (once the root is fixed on an hexagonal face) and show how in all cases the tree can be shortened by repeated applications of the irreducible ring-exchange terms in Fig. 15. A visual summary of all the cases is displayed in Fig. 16. In the list below we refer to $A$ as being a node at depth $h-1$ parenting a node at the bottom of the tree, at depth $h$.

1. $A$ has 6 children. In this case $A$ must be centred on a hexagonal face. Since no node can have more than 6 children, $A$ can have no parent node, and one can apply $H_8$ to truncate the tree.

2. $A$ has 4 children (the lattice forbids the possibility of 5 children). This vertex must be centred on a hexagonal face. Apply $H_8$ to shorten the tree.

3. $A$ has 3 children. This vertex must be centred on a hexagonal face (placing it on a triangular face yields a cycle of length 15, and any cycle emerging from two dimer configurations must be of even length, so this contradicts the initial assumption). Depending on the location of the children of $A$, one can always apply either $H_5$, $H_6$, or $H_7$.

4. $A$ has 2 children and can be centred on either a hexagonal or on a triangular face. If it is centred on a triangular face, one can apply $H_1$ on two neighbouring hexagons. If it is centred on a hexagonal face, one can always apply either $H_2$, $H_3$, or $H_4$.

5. $A$ has 1 child and can be centred on either a hexagonal or on a triangular face. If it is centred on a triangular face, one can apply $H_1$. If it is centred on a hexagonal face, one can always apply either $H_2$, $H_3$, or $H_4$.

This iterative procedure continues until termination on the hexagonal-faced root, where one can apply either of $H_1$-$H_8$ to transform $C$ into a collection of trivial cycles. $\qquad\square$

*Proof.* The explicit proof of Lemma 2.2 is done with induction on the number of vertices lying on a cycle $C$ enclosing only trivial cycles, following the logic outlined in Sec. 2.4. We label vertices enclosed by the cycle $C$ according to Fig. 17. By exhaustion of dimer configurations

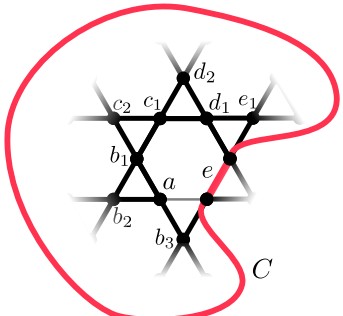

Figure 17: A cycle $C$ in $M_1 \cup M_2$, in which $e$ is a dimer from $M_1$, enclosing only trivial cycles and a vertex $a$ neighbouring the cycle. By exhaustion of dimer configurations in vicinity of $a$, it is shown that $C$ can always be extended to contain $a$.

in the vicinity of vertex $a$ it is shown that one of the moves $H_1$-$H_7$ always can be applied to include $a$ in $C$ and hence increase the number of vertices on the enclosing cycle $C$:

1. If $ab_1 \in M_1$:

   (a) If $c_1c_2 \in M_1$: either $d_1d_2 \in M_1$ and $H_2$ can be applied, or $d_1e_1 \in M_1$ and $H_3$ can be applied.

     (b) Else if $c_1d_1 \in M_1$: apply $H_1$.

     (c) Else if $c_1d_2 \in M_1$, then $d_1e_1 \in M_1$: apply $H_2$.

2. Else if $ab_2 \in M_1$:

     (a) If $b_1c_2 \in M_1$: either $c_1d_2 \in M_1$ (and then $d_1e_1 \in M_1$) and $H_5$ can be applied, or $c_1d_1 \in M_1$ and $H_2$ can be applied.

     (b) Else if $b_1c_1 \in M_1$: either $d_1d_2 \in M_1$ and $H_3$ can be applied, or $d_1e_1 \in M_1$ and $H_4$ can be applied.

3. Else if $ab_3 \in M_1$:

     (a) If $b_1b_2 \in M_1$: if $c_1c_2 \in M_1$, then either $d_1d_2 \in M_1$ and $H_5$ can be applied, or $d_1e_1 \in M_1$ and $H_6$ can be applied. Else if $c_1d_2 \in M_1$ (and then $d_1e_1 \in M_1$), then apply $H_7$. Else if $c_1d_1 \in M_1$: apply $H_2$.

     (b) Else if $b_1c_2 \in M_1$: either $c_1d_2 \in M_1$ (and then $d_1e_1 \in M_1$) and $H_6$ can be applied, or $c_1d_1 \in M_1$ and $H_3$ can be applied.

     (c) Else if $b_1c_1 \in M_1$: either $d_1d_2 \in M_1$ and $H_4$ can be applied, or $d_1e_1 \in M_1$ and $H_3$ can be applied.

$\square$

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
