# Peer review of "Ergodic Archimedean dimers"

_SciPost Physics_

## Round 1 · Referee Report · Anonymous (Referee 1) · 2023-6-12

Strengths

1- Systematic study of ergodicity in quantum dimer models on all 2d Archimedean lattices.

Weaknesses

1- Some of the results are already partially discussed elsewhere
2- The minimal set of local moves are not found for all lattices

Report

In this paper, the authors study dimer coverings (perfect matching) on the eleven possible 2d Archimedean lattices. They show that any dimer covering can be transformed into any other covering using a series of explicit local moves. For five of those lattices they further demonstrate that none of the local moves used are superfluous, while leaving that possibility open for the remaining six. This study is motivated by the issue of ergodicity in quantum dimer models, especially at their Rokshar-Kivelson points.

Overall the paper is interesting. It is also the first one trying to perform a more systematic study of this question, and it improves on several results on specific lattices which were briefly studied before. I also think the results are correct, and the arguments presented throughout are convincing.

While this definitely deserves publication in some form, it might be more suited for publication in Scipost Physics Core rather than Scipost Physics. Especially since this note, as the authors themselves write, is a reasonably short read and uses proof strategies which may be found elsewhere in the literature. Another reason is that it does not fully solve the question of a minimal set of local moves, which is an interesting one.

I have a few minor remarks:

a) Pages 6, second paragraph: replace On triangular lattice by On the triangular lattice.

b) Page 7. The discussion in the second paragraph could be improved by mentioning what happens to the phase with $\mathbb{Z}_2$ symmetry in the bipartite case. Also, dimer-dimer correlations do not decay algebraically on all bipartite lattices.

c) In the conclusion, a possible extension to quasicrystals in mentioned. However in Reference 46 a finite density of momomers appears unavoidable. How would this fit with the approach advocated in the present paper?

d) Also in the conclusion, the fact that certain materials realize some of the lattices investigated in this paper does not necessarily mean quantum dimer models are relevant for the study of such materials.

  • validity: high
  • significance: good
  • originality: good
  • clarity: high
  • formatting: excellent
  • grammar: good

Author:  Zhao Zhang  on 2023-06-23  [id 3758]

(in reply to Report 1 on 2023-06-12)

We thank the referee for carefully reviewing our work and providing helpful and accurate evaluations. Each of the remark from the referee is addressed in our resubmitted manuscript and responded to as below.

a,c,d: We thank the referee for pointing out these inaccuracies. We have rectified the text surrounding the relevant sentences in the revised manuscript. Modified text is shown in red in the attached version of the manuscript.

b: We have now specified that the quoted result holds for the square lattice (and not generally for bipartite lattices). For our own education we ask the referee to kindly point us to relevant references showing that the dimer-dimer correlations do not decay algebraically on some bipartite lattices.

---

## Round 1 · Referee Report · Anonymous (Referee 2) · 2023-6-18

Strengths

1- Interesting and useful results concerning the ergodicity of dimer coverings on archimedean lattices in terms of simple few-dimer moves.

Weaknesses

1- A large fraction of the results follow from a direct application of the the method introduced Kenyon and Rémila [Ref 25 of the manuscript]. More generally there is some significant overlap with previous works.

Report

This paper is well written and presents interesting results concerning the ergodicity of dimer coverings on Archimedean lattices in terms of simple few-dimer moves. Among applications of these results the authors discussed the ground-state of quantum dimer models at so-called 'RK' points. The method to prove the ergodicty of a given set of moves is directly inspired from Ref. [25].

I think that this work cannot be considered as a breakthrough or something opening a new pathway in the field. So, instead of ScipostPhysics I recommend this paper for publication in ScipostPhsyics Core, after the authors have considered the suggested changes below.

Requested changes

1- "Moreover, the statistical properties of closed-packed dimer coverings appear in quantum dimer models at the critical point at which such models become tractable." RK points are not necessarily critical points (examples: triangular QDM or the exactly solvable kagome lattice QDM).

2- "a sum of projectors and the unique ground state is a uniform superposition of the classical dimer configurations". Depending on the boundary conditions the ground state is not always unique.

3- "In both these cases, a single ring-exchange dimer move, acting on the fundamental square and hexagonal plaquettes, respectively, is sufficient to ensure ergodicity, see Fig. 1." To be more precise the authors might mention already here the existence of 'staggered' configurations (with maximum height tilt), which are mentioned only later in the text.

4- About the necessity of the L moves on the triangular lattice: "On triangular lattice, the necessity of the L and the B move can be shown by appealing to particular configurations that only have these respective tiles flippable. For instance, the columnar configuration, see the left panel of Fig. 5, proves the necessity of the L move." I am confused by the fact that other moves (involving several tiles) are in fact possible in the configuration the left panel of Fig. 5. For instance, some loops of length 8 are also flippable. Similarly, in the Fig. 6 one can also observe flippable loops of length 10 in the (4,6,12) lattice. Since the necessity of some specific moves is an important result in this study the authors should clarify the argument.

5- Concerning the special case of the kagome lattice, the so-called 'pseudo-spin representation' (Ref [34]) seems to be an economical way to prove the ergodicity of the H1-H8 moves (Fig. 8). This could mentioned, at least in appendix C.

6- Again on the kagome lattice: "In fact, a Hamiltonian consisting only of the off-diagonal terms is exactly solvable, since all operators in the Hamiltonian commute with each other". The authors should be more precise here when specifying which parts of the Hamiltonian are commuting, since the off-diagonal terms associated to the different moves H1-H8 do not commute with each other. The operators which indeed commute are the sums of these terms associated to a given hexagon.

  • validity: high
  • significance: good
  • originality: good
  • clarity: high
  • formatting: excellent
  • grammar: excellent

Author:  Zhao Zhang  on 2023-06-23  [id 3757]

(in reply to Report 2 on 2023-06-18)

We thank the referee for carefully reviewing our work and providing helpful and accurate evaluations. Each remark of the referee has been addressed in our resubmitted manuscript and responded to as below.

1,2,3,5,6: We thank the referee for spotting inaccuracies and suggesting helpful changes. We have incorporated the suggested changes in the resubmitted manuscript, highlighted in red.

4: First, we clarify that by considering ring-exchange moves along what we define to be the irreducible even-length cycles of the lattice (as defined below Fig. 3), we are guaranteed that any other ring-exchange term one can think of can be decomposed into those considered. The columnar dimer configuration in the left panel of Fig. 5 does not have flippable tiles corresponding to T moves (of either two orientations) or B moves (of either three orientations). Presumably, the referee is referring to loops of length 8 involving three Lozenges glued together. Such a move can be expressed as compositions of Lozenge moves and is hence redundant. One is of course free to include further terms in the candidate set to begin with (which could affect the counter examples we construct to prove necessity), but this would render the set non-minimal. Regarding the length-10 flippable tiles on the (4,6,12) lattice, which if we understand the referee correctly involves one hexagon and two squares, we refer to the above reply: within our set of three candidate moves associated with irreducible cycles, the length 10-move move is redundant and can be expressed as a composition of the square flip move (Fig. 6) and the hexagon flip move (Fig. 10). We have added a clarifying comment on this in the first paragraph of Sec. 3.

---

## Editorial Decision

resubmitted